# Importance of Virulence Factors for the Persistence of Oral Bacteria in the Inflamed Gingival Crevice and in the Pathogenesis of Periodontal Disease

**DOI:** 10.3390/jcm8091339

**Published:** 2019-08-29

**Authors:** Gunnar Dahlen, Amina Basic, Johan Bylund

**Affiliations:** Department of Oral Microbiology and Immunology, Institute of Odontology, Sahlgrenska Academy, University of Gothenburg, SE-40530 Gothenburg, Sweden

**Keywords:** periodontal disease, host response, infection, inflammation, oral microbiota, virulence factors, metabolites

## Abstract

Periodontitis is a chronic inflammation that develops due to a destructive tissue response to prolonged inflammation and a disturbed homeostasis (dysbiosis) in the interplay between the microorganisms of the dental biofilm and the host. The infectious nature of the microbes associated with periodontitis is unclear, as is the role of specific bacterial species and virulence factors that interfere with the host defense and tissue repair. This review highlights the impact of classical virulence factors, such as exotoxins, endotoxins, fimbriae and capsule, but also aims to emphasize the often-neglected cascade of metabolic products (e.g., those generated by anaerobic and proteolytic metabolism) that are produced by the bacterial phenotypes that survive and thrive in deep, inflamed periodontal pockets. This metabolic activity of the microbes aggravates the inflammatory response from a low-grade physiologic (homeostatic) inflammation (i.e., gingivitis) into more destructive or tissue remodeling processes in periodontitis. That bacteria associated with periodontitis are linked with a number of systemic diseases of importance in clinical medicine is highlighted and exemplified with rheumatoid arthritis, The unclear significance of a number of potential “virulence factors” that contribute to the pathogenicity of specific bacterial species in the complex biofilm–host interaction clinically is discussed in this review.

## 1. Introduction

Periodontal diseases affect the supporting tissues of teeth. The most common, gingivitis and periodontitis, are inflammatory diseases that are induced and maintained by the polymicrobial biofilm (dental plaque) that are formed on teeth in the absence of daily oral hygiene procedures. While gingivitis is a reversible inflammatory response without loss of bone support, periodontitis includes the destruction of the periodontal attachment and the alveolar bone. Peri-implantitis is the term used for a similar inflammatory reaction as periodontitis, but around dental implants, also here including the loss of bone support [1].

Periodontitis is the result of a complex interplay between microorganisms of the dental biofilm and the host. The role of specific microorganisms and their products in the disease initiation and propagation is still unclear [2,3]. The severity of the periodontal disease also depends on environmental (e.g., smoking) and host risk factors (for example genetic susceptibility) [4]. Lately, numerous studies have shown associations between periodontal disease and a number of systemic diseases, such as cardiovascular disease, diabetes mellitus, Alzheimer’s disease, and rheumatoid arthritis [5]. This has intensified the research on the role of the microorganisms and their virulence factors in periodontitis. The purpose of this review is to high-light the complexity of the host-microbe relationship in periodontitis as well as the capacity of ordinary low virulent oral commensals to adapt and survive in the periodontal pocket, and to become infectious and contribute to systemic effects on the host.

## 2. The Role of Micro-organisms in Periodontitis—A Historical Perspective

The paradigm that dental plaque was the cause of gingivitis was established in the “gingivitis in man study” [6], where, in an experimental approach, a group of volunteers abstained from oral hygiene procedures resulting in plaque accumulation and gingivitis development. When oral hygiene procedures were reinstalled, the gingivitis was resolved within a week. The interpretation that followed was that the dental plaque induced the gingival inflammation and, if left untreated, this would inevitably lead to periodontitis [3]. Plaque was thought to cause periodontitis, and therefore plaque control became the cornerstone in the treatment of periodontal disease, and still is. Since the periodontitis distribution in the population was skewed, focus was next directed to specific bacteria, present in the dental biofilm of gingival/periodontal pockets and were more or less associated with periodontitis. Specific microorganisms, termed “putative periodontal pathogens”, were hypothesized to be responsible for gingival tissue breakdown and the disease progression [7,8]. Even if Koch’s postulate was found to be inadequate to apply to complex diseases like periodontitis, additional criteria for the elucidation of pathogens capable of causing periodontitis were suggested [9]. This bacteria-centered view of the disease led to the concept of an infectious nature of periodontitis, where treatment was aimed at eliminating putative pathogens e.g., with antibiotics [10].

It soon became clear that bacteria in the polymicrobial biofilm were necessary but not sufficient to explain why some individuals developed periodontitis while others did not, despite a similar composition of the subgingival microbiota. The existence of refractory cases that responded poorly to treatment and in which the disease continued to progress despite comprehensive periodontal treatment directed the focus towards the host and the inflammatory response—the disease susceptibility model [3,11,12]. This host-centered approach in search for high-risk groups and individuals discovered that diabetes patients or smokers were overrepresented among those with severe periodontitis [13,14]. The focus was on the host response and search for genetic grounds became intensive based on family pattern and twin studies of periodontitis [15,16]. The general epidemiological pattern, comprising of approximately 10–12% of the populations worldwide, irrespective of hygiene level and access to dental treatment, suffering from severe periodontitis with the risk of tooth loss [17] indicates that subgroups with genetic susceptibility for severe periodontitis exist in all populations. In a recent review it was concluded that up to one third of the variance of periodontitis is due to genetic factors [16]. The search for specific genes or gene polymorphism to explain the genetic role in periodontitis have so far been only moderately successful but the heritability of the disease is extremely complex and likely also influenced by epigenetic mechanisms [18,19].

While bacteria have a clear role in the periodontal disease aetiology, the relationship to other microorganisms, such as yeasts, is more uncertain [20]. Herpesviruses and Epstein-Barr virus, on the other hand, have been more strongly implicated in periodontitis and a link has also been suggested between periodontal herpesviruses and systemic diseases [21].

Nowadays periodontitis is described as an inflammatory disease induced and maintained by the polymicrobial biofilm formed on teeth, based on the polymicrobial synergy and dysbiosis hypothesis [22,23,24]. This hypothesis implies that the balance (homeostasis) between the microorganisms in the dental biofilm and the host response is disrupted due to fluctuations and burst of activity of the microorganisms or due to an imbalanced host response. This imbalanced condition in which the normal microbiome structure is disturbed is termed dysbiosis [24]. The magnitude of the inflammatory response is host related (susceptibility) and host related factors are responsible for the disease progression. The role of the classic, putative periodontal pathogens is still unclear and the infectious nature of the disease has been depreciated [23,24,25]. However, the enormous complexity and variability that takes place within the dental biofilm community and the similar complexity and variability in the inflammatory response is challenging. The role of the microorganisms in periodontitis is poorly understood and the many hypothesis launched are still hypotheses.

## 3. “Putative Periodontal Pathogens”

*Aggregatibacter actinomycetemcomitans* (previously *Actinobacillus actinomycetemcomitans*) was discovered to be closely related to localized forms of periodontal disease in young individuals. This discovery made it plausible that some microorganisms were more important for periodontal disease development than others and such microorganisms were termed “putative periodontal pathogens” [25,26]. Later, Socransky et al. [27] grouped three species (*Porphyromonas gingivalis, Tannerella forsythia* and *Treponema denticola*) to the so-called red complex, which was statistically strongly associated with periodontitis (Table 1). Another group including species such as *Prevotella intermedia*, *Fusobacterium nucleatum*, *Campylobacter rectus* and *Campylobacter gracilis* were associated with periodontitis to a milder degree and these were termed the orange complex [27]. Notably, most of these putative periodontopathogens are Gram-negative, strictly anaerobic species (except for *A. acinomyctemcomitans*), as well as proteolytic and thus well adapted to the inflamed periodontal pocket (Table 1). Occasional Gram-positive species, e.g., *Parvimonas micra* (formerly *Peptostreptococcus micros*) were included in the orange complex [27].

While some species, such as *A. actinomyctemcomitans* and *P. gingivalis,* are extremely well studied and described in several recent reviews, e.g., [28,29,30], phenotypical characterization of other bacterial species is more limited. Spirochetes are seriously underestimated, although they have been known to predominate the deep periodontal pocket from microscopic studies in the 1970s [31]. Studies using molecular biology methods have disclosed a number of unculturable *Treponema* genotypes, but their phenotypic characteristics are largely unknown [32,33].

It should also be noted that the concept of “species” is a man-made distinction and most species can be further specified into genotypic and/or phenotypic subtypes based on defined criteria. Clearly certain subtypes are more associated to oral infections than others, but unfortunately, the identification of bacterial species to a subtype level is rare in clinical oral microbiology studies. The best-known example of a specific bacterial subtype with implications for periodontal disease is the JP2 genotype of *A. actinomycetemcomitans,* which is a high virulent clone (high toxic clone) strongly associated with severe periodontal breakdown in young individuals of West African populations [34].

The list of “putative periodontal pathogens”, or bacterial species associated to periodontitis, has gradually expanded and included 17 different species in a recent review [35]. Along with the use of more sensitive detection methods such as NGS (Next generation sequencing) the number periodontitis associated microorganisms have expanded further but their role in the pathogenesis of periodontitis remains elusive [36,37]. It is possible that certain microorganisms are more important for periodontal disease progression than others, but that can only be ruled out in prospective longitudinal studies over years without intervention, and very few such studies have been conducted [38].

## 4. Periodontitis—An Inflammatory Disease or an Infectious Disease?

Inflammation is a host tissue response to an assault commonly triggered by microorganisms (or other stimuli such as chemicals, radiation and trauma) and their products (metabolites, endotoxins) released from them. Clinically, inflammation often relates to some kind of pathology, but it is important to remember that inflammatory reactions are absolutely critical for our well-being and the process has evolved to provide rapid and early protection. We likely experience thousands of small clinically ‘invisible’ inflammatory reactions along the mucosal membranes every day as a result of various assaults involving microorganisms, without considering these inflammatory responses as infections (or even disease). These cases could not be considered pathologic and represent situations where the inflammatory response is physiologic and provide early protection from potentially dangerous events (see below).

There are many definitions of infection, most of them aimed at primary pathogens and specific infections, while it is more controversial to find a definition that fits to infections involving commensals [39]. In line with the “damage-response framework” [39], which can be applied to various oral infections, the term infection can be defined as ”the process in which pathogenic organisms (ambionts) invade the tissues or organs of the body and cause injury (damage) followed by reactive phenomena” according to Dorland´s Medical Dictionary [40,41]. The critical issue here is invasion (further discussed below) and ordinary commensals are not part of any infection as long as they are colonizing (present) on the external side of the epithelial lining (barrier). That is a ‘normal’ condition. This means that in a state of commensalism and colonization (such as the subgingival dental plaque in health or gingivitis) a homeostatic balance between the microbiota and the host response (a homeostatic inflammation) is created, according to Lamont et al. [23]. This controlled immune-inflammatory state where the inflammatory response of the host causes no apparent damage should not be considered as pathological but rather as a normal response (a physiological inflammation). Thus, gingival inflammation as a response to the dental biofilm is a state of normality developed through the evolution and present in almost 100% of adults of all populations worldwide [24,42].

Infection occurs when the virulence, the number, and the exposed time supersede the local and general host’s defense, which leads to a pathological reaction in the host’s tissues [43]. Virulence is the relative ability of a microorganism to cause disease and consequently, virulence is a microbial property that can only be expressed in a susceptible host. Hence, virulence is not an independent microbial property, because it cannot be defined independently of a host [39]. In that perspective all bacteria in the dental biofilm may contribute to the host response and it is not meaningful to distinguish pathogenic microbes from non-pathogenic at that stage (gingivitis). The dental biofilm is the natural habit for these microorganisms and the change in microbial composition that occurs over time may just be a consequence of the changing environment brought about by the host (inflammatory) response. This implies that bacterial species that are better adapted to the changing environment (deeper pocket, anaerobiosis, mechanical retention, alkaline environment, and increased exudate flow with an abundance of serum proteins, complement factors, neutrophils, lysozyme) will thrive and thereby increasingly (or at least consistently) fuel the inflammatory response. This may occur without bacterial invasion and also without necessarily causing any apparent damage. Therefore, gingivitis is not defined as an infection.

In certain periodontal conditions aside from chronic periodontitis, the case for an infectious nature of the problem is strong, e.g., exacerbations and abscess formation due to trauma or blockage of drainage by, e.g., calculus, fillings, or prosthesis. Periodontal infections resemble other dentoalveolar infections regarding their microbial composition and progression [44]. They are endogenic polymicrobial, predominantly anaerobic infections that are characterized by a destroyed internal regulation in the biofilm community with increased activity and growth of bacteria (burst) originating from the dental biofilm. In addition, periodontal infections usually feature bacterial invasion which requires detachment from the plaque. One other condition where bacterial invasion is rather obvious is peri-implantitis which is an endogenic polymicrobial anaerobic infection where the bacteria originate from the biofilm on the implant surface and with a microbiota which resembles that of the periodontal pocket [45]. Due to the anatomical difference from teeth and lack of an intact epithelial barrier, bacterial invasion can be facilitated in peri-implantitis. Clinically visible pus formation and increased number of neutrophils found in biopsies support the view of peri-implantitis as an infectious process [46].

In these conditions mentioned above (periodontal infections), the infections are polymicrobial and although some bacterial species are more frequently found (and may even play an important role), it is not possible to exclude any of the microorganisms from taking part in the “infection” and to solely act as innocent bystanders [2]. Numerous studies have been performed to explain virulence mechanisms in vitro but in accordance with the above definition of virulence this cannot be fully evaluated for various microorganisms independently of a host. However, experimental infections in animals have clearly shown that mixed infections are more infectious than mono-infections, and have also emphasized the importance of anaerobes without the necessity to include “putative pathogens” in such infections [47].

In contrast to periodontal infections, chronic periodontitis is a pathological inflammatory disease, as a response to prolonged (months, years) exposure to, and not infected by a polymicrobial community in the gingival/periodontal pocket. The tissue responses involve alternating destructive phases and repair or healing phases, but with a net loss of attachment and bone. The microorganisms are necessary components but not sufficient to explain the tissue damage. In view of the “damage–response framework” as discussed earlier, both the host and microbes contribute to pathogenesis. As mentioned, the role of specific microorganisms and certain virulence mechanisms in triggering this pathologic inflammation remain unclear. Most factors that are commonly discussed in terms of “virulence factors” among the putative periodontal pathogens should rather be termed “survival factors” since they do not necessarily constitute factors for pathogenicity and damage but for the living, growth and survival of the bacteria in deep, inflamed periodontal pockets.

## 5. Factors that Promote Bacterial Colonization and Persistence

The oral microbiome in humans is distinctly different from that of other species. Similarly, the human oral microbiome is distinctly different from that of other body compartments, such as the skin, intestine, and vagina [48]. The oral microbiome comprises a highly diverse microbial population, involving more than 700 species [49]. Furthermore, within the oral microbiome, the dental biofilm has its own microbiome characterized by strong tooth surface adhering streptococci and Actinomyces [50]. Adhesion is essential for colonisation within the oral cavity, and it is regulated primarily by the host and host receptors on the mucosal surface and teeth. In fact, the host selectively, very early after birth, allows the microorganisms that best fit to the receptors of each individual and the specific environment of the oral cavity to colonize. The receptor interaction is mediated by microbial adhesins, which are surface structures such as fimbriae, capsule, lipopolysaccharides and others.

In the area of the gingival crevice, the microenvironment is different from other parts of the tooth surface, with the major nutrition probably coming from gingival crevicular fluid (GCF). Since GCF, in contrast to saliva, does not contain any sugars, the main source of nutrition for the microbiota in this niche are the proteins. Hence, the main metabolic pathway is proteolytic, thus favouring the proteolytic rather than the saccharolytic microorganisms. In addition, the GCF, which is a serum exudate, also contains a number of growth supporting factors such as vitamins (e.g., K-vitamin or menadione), hormones (e.g., oestrogen) and specific serum proteins/peptides (e.g., hemin) all favouring many of the fastidious Gram-negative anaerobes that adapt and grow concomitantly with gingival inflammation and deepening gingival pockets [24,50,51].

Gingival inflammation typically results in a deepening of the gingival pocket owing to swelling, oedema and resulting in increased flow of GCF. This has an impact on the type of colonizing and growing bacteria since it further provides nutritious advantages to proteolytic species. Furthermore, the lowering redox potential (Eh), which is arising from the swelling, favours the anaerobes. In contrast to the supragingival plaque, the adhesion of the microorganisms does not play a crucial role in the gingival pocket and motile bacteria (*Treponema, Campylobacter, Selenomonas* spp.) are able to establish themselves by mechanical retention [50]. The GCF contains humoral defence factors (antibodies, complement, antimicrobial peptides) as well as inflammatory cells such as neutrophils and monocytes. This selects for bacteria that can escape phagocytosis and killing by producing anti-phagocytic capsules (*P. gingivalis*) or leukotoxins (*A. actinomycetemcomitans*), or that simply are proteolytic enough to degrade most proteins, including humoral antimicrobial factors such as immunoglobulins (IgG), complement factors, or antimicrobial host defence peptides [52]. The exudate also contains lysozyme, an enzyme directed towards the peptidoglycan of the bacterial cell wall, which is protected by the outer membrane of Gram-negative microorganisms (Gram-positives lack such outer membrane). Consequently, Gram-negatives appear to have a higher survival rate in the inflamed gingival pocket. Inevitably, inflamed gingival pockets contain a microbiota dominated by Gram-negative, anaerobic, proteolytic, and motile bacteria as they are favoured by and adapted to the changing pocket environment (Table 1). This character of the subgingival plaque microbiota has been repeatedly shown in numerous studies for decades using microscopic-, culture- and molecular biology methods [8,53,54,55,56]. The composition of the subgingival microbiota can thus convincingly be explained as a result of a changing ecology involving only commensals (essentially non-pathogenic microorganisms) that are adapting to the new environment [8].

## 6. Factors Promoting Gingival Inflammation

### 6.1. Bacterial Metabolites as Pro-Inflammatory and/or Cytotoxic Agents

The products of bacterial metabolic processes, the metabolites, can be used by the bacteria for the construction of new macromolecules, but may also remain by-products with cytotoxic potential and take part in the host-microbe interplay. In a combined metagenome/metatranscriptome analysis that compared active and non-progressing sites in periodontitis patients it was confirmed that proteolysis is associated with progressing periodontal sites [57]. Such bacterial metabolism results in the formation of various compounds, some of which are suggested to participate in the pathogenesis of periodontitis [58].

Bacterial hydrogen sulfide (H_2_S) is formed predominantly by degradation of the amino acid L-cysteine or the peptide glutathione by various bacterial species that are associated with periodontitis [59]. H_2_S is also formed from sulfate by sulfate reducing bacteria. It is a volatile sulfur compound that has been shown to induce the secretion of the pro-inflammatory cytokines IL-1β and IL-18 in monocytes in vitro through the formation of the NLRP3 inflammasome (Figure 1) [59]. These and other pro-inflammatory cytokines are necessary to sustain or even strengthen the inflammatory reactions in the gingiva. Furthermore, H_2_S can induce apoptosis in human gingival fibroblasts [60] and split disulfide bonds in host proteins. These mechanisms, that in different ways trigger inflammatory reactions, are proposed to contribute to the disruption of homeostasis between host cells and bacteria, and to participate in the development of disease. In support of this notion, the sulfur compound metabolic processes have been shown to be enhanced in periodontitis progressing sites [57]. Other volatile sulfur compounds, such as methyl mercaptan, dimethyl disulfide, and dimethyl sulfide, are formed from the degradation of L-methionine among other metabolic pathways, and have similarly been suggested to participate in the host-microbiome crosstalk during the induction and progression of periodontal disease [61].

Additional products of the catabolism of L-cysteine and L-methionine are, together with H_2_S and methyl mercaptan, ammonia and pyruvate. Ammonia (NH_3_), a product of numerous metabolic processes that take place in the periodontal pocket including the transformation of arginine or lysine [62], is suggested to, apart from its effects on pH, also affect neutrophil function [63]. Apart from ammonia, the degradation of arginine can also, result in the by-product and amino acid citrulline [62]. Pyruvate can be degraded into acetic acid or formic acid for energy production [64].

Various carboxylic acids, i.e., acetic acid, butyric acid, formic acid, isobutyric acid, isovaleric acid, lactic acid, propionic acid, phenylacetic acid, succinic acid, and valeric acid are by-products of the deamination of amino acids to generate energy. The presence of many of these have been found to be elevated at inflamed and diseased periodontal sites [65,66,67,68]. Comparably to other metabolites, also these mainly short-chain fatty acids have been suggested to act as potent and modifying factors in periodontitis. Butyric acid (Table 1) has for instance been shown to be capable of inducing apoptosis and autophagic cell death in gingival epithelial cells [69], to induce apoptosis in inflamed fibroblasts [70], to induce ROS production, and to impair the cell growth of gingival fibroblasts [71]. These effects stimulate inflammation and are thus, thought to contribute to initiation and the prolongation of periodontitis [66]. Interestingly, many of these metabolites are recognized by specific free fatty acid receptors (FFAR) on leukocytes such as neutrophils and potently attract and activate these cells [72]. This receptor group all bind short-chain fatty acids and include FFA1R (earlier known as GPR40), FFA2R (earlier known as GPR43) and FFA3R (earlier known as GPR41). Such recognition of free fatty acids likely influences both metabolic and immunologic processes [73], and although there has been quite some interest on how free fatty acids produced by the gut microbiota affect health and disease [74], to our knowledge, nothing is known regarding oral bacteria and whether free fatty acid production has bearing on inflammatory reactions in the gingiva.

Apart from the formation of new macromolecules and the interplay with the host, the metabolites also contribute to environmental changes that provide the transformations that are needed for the bacterial shift from homeostasis. Sulfur compounds are, for instance, reducing agents that contribute to lowering the redox potential which favor obligately anaerobic bacteria. Similarly, ammonia is an advocated contributor to the slightly alkaline pH of the periodontal pocket. Other metabolites that have been discussed in the literature as bacterial cytotoxic end-products playing a mediating role in periodontal disease pathogenesis include indole, amines (derivatives of ammonia), and the polyamine cadaverine, which is produced from lysine and shown to be elevated at diseased sites [75].

### 6.2. Bacterial Cell-Wall Constituents as Pro-inflammatory Factors

Bacterial cell-wall constituents such as short and long fimbriae, outer membrane vesicles and endotoxin (lipopolysaccharide; LPS) have been proposed in various ways to interact with (or even to cause subversion of) the inflammatory response [76,77]. Most attention for immune modulation has been directed toward LPS that builds up the outer membrane of all Gram-negative bacteria and consists of a Lipid A moiety, a core polysaccharide, and a polysaccharide chain of repeating sugar subunits. The latter is an important antigen (O-antigen) that has been used for classification of several enteric genera. LPS is a very immunoreactive molecule and seen as a molecular pattern recognized by potential host organisms—genome-encoded receptors capable of reacting to LPS are found in organisms ranging from plants to mammals [78]. In humans, LPS is sensed primarily by toll-like receptor (TLR) 4, which is expressed by immune cells, but also by a wide variety of other cell types. When bound by LPS, TLR4 activates the pro-inflammatory transcription factor NF-κB that enters the nucleus and initiates gene transcription (Figure 2).

This is a pivotal event that regulates the production of multiple pro-inflammatory cytokines [78]. The broad pro-inflammatory response triggered by NF-κB makes LPS a primary candidate for immune-modulation of gingival inflammation. Since all Gram-negative bacteria release LPS, they may all be involved in triggering/sustaining periodontal inflammation. However, the potency of LPS to trigger inflammation varies among different bacteria, and although this has been studied extensively for *P. gingivalis* [79], less is known about the bioactivity of LPS from other Gram-negatives residing in inflamed gingival pockets.

## 7. Invasion and Factors that Promote Tissue Degradation

As mentioned above, bacterial invasion is a key process in order to define infections by commensal bacteria, and although invasion seems well established for, e.g., peri-implantitis, it is less clear if it occurs and if it is of importance for chronic periodontitis. How bacteria penetrate through the epithelium and reach the gingival connective tissues is also unclear [80,81], but the pathways commonly discussed [81,82] are: the intercellular route, the intracellular route and by trauma. 

### 7.1. The Intercellular Route

An intercellular route is suggested to take place primarily by motile species, e.g., spirochetes [82]. The junctional epithelium is non-keratinized and thin (4–5 cells thick), and in a state of inflammation, the cells are not tightly joined in order to facilitate gingival exudate and migration of neutrophils and monocytes through the gingival barrier. It is hypothesized that this intercellular passage also allows motile bacteria such as *Treponema* and *Campylobacter* species to penetrate the barrier. *Treponema* spp. and *P. gingivalis* produce specific gingipains that can degrade epithelial junctional proteins (E-cadherin and occludin), and this impairs the junction-related structures [83,84].

Invasion is more likely to take place if the epithelial barrier (junctional epithelium) is disrupted in the case of ulceration. In acute necrotizing ulcerative gingivitis (ANUG) spirochetes are shown to invade the underlying connective tissue by motility [85]. A similar situation is also seen in peri-implantitis where the epithelium fails to cover the connective tissues of the apical part of the peri-implant pocket [46,86]. The periodontal abscess is another example of invasion into the tissues by bacterial growth, and the abscess can sometimes even lead to fistula formation. Invasion in these examples is due to bacterial multiplication and expansion, and it is associated with heavy neutrophil attraction and suppuration, and sometimes symptoms characteristic of an acute infection.

### 7.2. The Intracellular Route

The intracellular route has gained more attention since it was noticed that viable and non-keratinized epithelial cells contain bacteria. Buccal epithelial cells regularly contain bacterial cells, mainly streptococci [87]. Interestingly, *P. gingivalis* and other periodontal bacteria have also been shown in such cells [88]. *T. forsythia, Prevotella intermedia* and *C. rectus* have been identified inside crevicular epithelial cells in vivo [89], and this uptake is mediated by receptor interaction between the bacteria and the epithelial cells [90]. Fimbriae (long and short fimbriae) is suggested to play a crucial role in bacterial invasion of cells and periodontal tissues through receptor interaction between the fimbriae and host cells [76]. *P. gingivalis* long fimbriae can be divided based on the *fimA* gene into 6 different subtypes of which *fimA* II and IV were more prevalent in periodontitis while *fimA* type I is more prevalent in healthy periodontal tissues [91]. Fimbriated *P. gingivalis* have been shown to be more efficient than fimbriae-deficient *P. gingivalis* to enter human dendritic cells in vitro [92], while Type II fimbriae were associated with increased proinflammatory and invasive activities in macrophages [76]. Interestingly, in one study a Type-1 fimbriae *P. gingivalis* strain induced more bone loss than a Type-II *P. gingivalis* strain in a mouse model [93]. Nevertheless, the importance of fimbriae for tissue invasion in the complex interaction between the subgingival microbiota and the host response in humans remains to be elucidated.

### 7.3. Invasion by Trauma

Bacterial invasion of the pocket epithelium and the underlying connective tissue in gingival biopsies from patients with periodontitis has been reported using various methods [81]. These studies clearly show presence of various periodontitis associated bacteria intracellularly in epithelial cells as well as in the connective tissue, but without explaining whether they have passed the barrier actively or accidentally by trauma (e.g., while taking the biopsies). Bacteria are commonly pushed into the connective tissues during manipulations and hygienic procedures (tooth brushing, flossing, tooth picks) and chewing. Dental hygienists or dentists, during cleaning and treatment procedures (depuration, scaling, ultrasonication, probing, periodontal surgery and tooth extractions), frequently cause the spread of microorganisms into the blood stream. Bacteremia during dental procedures is thus a well-known phenomenon, and poor oral hygiene, with an abundance of plaque and inflammation, increases the likelihood of bacterial spread out in the tissues. It is likely that most people have bacteremia daily without any (severe) consequences but it is also possible that bacteria can survive within the periodontal tissues and not being immediately eliminated by the host defense.

### 7.4. Tissue Degrading Factors

Tissue degrading or histolytic enzymes produced by bacteria were previously thought to be directly responsible for periodontal tissue breakdown [94], comparably to similar enzymes of pathogenic tissue invading bacteria such as *Clostridium histolyticum, Staphylococcus aureus* and *Streptococcus pyogenes* (Group A streptococci) [94]. Hyaluronidase (“spreading factor”), chondroitin sulphatase, and beta-glucuronidase produced by various oral bacteria such as streptococci, peptostreptococci, and corynebacteria (diphteroids) were thought to facilitate the spread of infections by degrading tissue components. Similarly, various bacterial proteases including collagenases were thought to play a major role by degrading tissue constituents in infections involving anaerobic proteolytic bacteria [94]. It was tempting to argue that collagenases and gingipains (previously termed “trypsin-like”) produced by several *Porphyromonas* species (e.g., *P. gingivalis* and *P. gulae*) degrading almost any protein in vitro also had a role in periodontal pathology. Although, realizing that host defense cells, e.g., neutrophils and macrophages, also produce similar enzymes, the possibility that bacterial enzymes contribute to tissue degradation has been diminished. 

## 8. Factors for Evasion of Host Defence

The host reacts extensively to the microbial challenge by employing the inflammatory reaction and assembling defense factors such as neutrophils, complement factors, and antibodies in the exudate and gingival pocket [95]. Microorganisms with the capacity to evade such host factors will increase their survival rate. As mentioned above, proteolytic enzymes may split or degrade the host’s defense molecules such as immunoglobulins and complement [96], but there are also more sophisticated mechanisms for the evasion of host defense.

### 8.1. Exotoxins

Toxins that destroy leukocytes, leukotoxins, are well-known virulence factors found in several pathogenic bacteria such as the PV-leucocidin in *Staphylococcus aureus* [97] and *Fusobacterium necrophorum* [98]. This explains the invasive and abscess-promoting character of these pathogens. Therefore, the leukotoxin-producing microorganism, *A. actinomycetemcomitans*, among the periodontal bacteria, has been extensively studied due to this particular factor [99,100]. The importance of leukotoxin in *A. actinomycetemcomitans* was emphasized when discovering the high toxic JP2 clone, which produces highly elevated levels of leukotoxin. Interestingly, Höglund et al. [101] correctly predicted that children of a Ghanaian cohort harboring *A. actinomycetemcomitans* and the high-toxic clone (JP2) in particular, at periodontal sites, had significantly more periodontal breakdown after 2 years compared with baseline, indicating a specific impact of the JP2 clone on the disease progression. The leukotoxin explains the specific ability of this bacterium to survive or escape the host’s defense (neutrophils present in the gingival pockets), but it does not explain the periodontal breakdown that is associated with this bacterium in aggressive forms of periodontitis in children and adolescents [102]. The fact that the toxin is both membrane-associated and secreted may be important since (at least) the secreted form would likely create a local niche free from viable neutrophils, which would benefit all bacteria present at this site.

### 8.2. Capsule Formation

The formation of an anti-phagocytic capsule is a well-known virulence strategy of many pathogenic microorganisms. *Streptococcus pneumoniae* and *Haemophilus influence* are exclusively dependent on capsule formation and vaccination of children towards their capsular antigens is today a reality [103,104]. Capsules are produced by many Gram-negative anaerobic rods, e.g., *Bacteroides, Prevotella,* and *Porphyromonas* [105,106]. The capsule is produced to escape phagocytosis and thereby intracellular killing by neutrophils and macrophages. The inability of the host’s defense to kill capsulated bacteria may result in spread of infections and to complications such as sepsis [107]. Six capsular serotypes have been identified among periodontal *P. gingivalis* isolates (Table 1). Two capsular types, K1 and K6, seem to include isolates with a lower adhesion capacity to human leukocytes in vitro than the other capsular types [105,108]. However, in vivo evidence supporting that any of the capsular types would be more associated with periodontitis than others are lacking.

## 9. Systemic Implications of Periodontal Bacteria

Bacteraemia and focal infections from the oral cavity have been associated with teeth and oral conditions for more than hundred years [109]. Formerly, the total extraction of all teeth was a common way to avoid future problems and full dentures was the only option for adults, and still is in many countries in the world. Incomplete endodontic treatment was thought to be the major cause of focal infections and apical periodontitis is still a suspected (by the medical profession) major point of entry for oral bacteria into the blood stream and further to various organs of the body. Today, the systemic effects from marginal periodontitis is considered to be much more important [109]. Not only bacteria as such but also products from subgingival bacteria (e.g., metabolites and endotoxins), and inflammatory mediators produced locally in the periodontal tissues, are being linked to the chronic inflammation seen in a number of systemic diseases such as atherosclerosis, diabetes, preterm birth, rheumatoid arthritis and Alzheimer’s disease [5,110]. Especially, the association between periodontitis and putative periodontal pathogens to Alzheimer’s disease has been in focus during the last decade and *P. gingivalis* LPS and outer-membrane vesicles have been suggested to be accumulated in brain tissues of patients with Alzheimer’s disease [77,111]. Most studies linking periodontitis to systemic diseases are association studies and it is less clear what (if any) unique contribution periodontitis might have that would not be found for other chronic inflammatory diseases in the body (e.g., in the gut).

However, one disease where the connection to periodontitis appears more solid than a mere association between the two diseases, is rheumatoid arthritis (RA), a widespread systemic inflammatory disease that is mainly manifested as peripheral polyarthritis (i.e., inflammation of peripheral joints) and that affects up to 1% of the global population [112]. RA is a heterogeneous disease but the majority of cases feature auto-antibodies directed towards citrullinated epitopes, so-called anti-citrullinated protein antibodies (ACPAs). Citrullination is a biochemical post-translational modification where arginine residues of peptides and proteins are converted into citrulline residues. In general, protein citrullination is of great physiological importance and endogenous citrullination is ascertained by a family of human enzymes known as peptidylarginine deiminases (PADs), some of which are primarily located immune cells, such as neutrophils. PAD-mediated citrullination of arginine residues may create neo-epitopes that could give rise to the development of ACPAs and one major research questions in RA deal with how, when, and where tolerance is broken to allow for the generation of ACPAs.

*A. acinomycetemcomitans* has been shown to trigger dysregulated citrullination of proteins, a process mediated by the activation and release of endogenous neutrophil PAD by the actions of the bacterial leukotoxin [113]. Although the pore-forming leukotoxin of *A. actinomycetemcomitans* is exclusive among oral bacteria, pore formation is by no means a unique feature of bacterial toxins in general. Additionally, *P. gingivalis* has been implicated as a potentially even more interesting link between periodontitis and RA when it was found that it expresses a bacterial PAD, typically referred to as PPAD, to distinguish it from the human isoforms [114]. PPAD seems to be rather uniquely expressed by *P. gingivalis* among the bacteria that interact with humans and it has the ability to create citrullinated neo-epitopes that are recognized by ACPAs from RA patients. Interestingly, *P. gingivalis* appears especially fit to do this, since the PPAD is specific for C-terminal arginine residues and such peptides are typically generated by the gingipains. Thus, these two *P. gingivalis* enzymes (PPAD and gingipains) appear to work in concert, in a way that maximize the levels of citrullinated epitopes which might thereby break tolerance and facilitate the generation of ACPAs [112].

## 10. Conclusions

Inflammation in the gingiva (gingivitis) is a normal host tissue response (a physiological inflammation) triggered by ordinary commensal microorganisms and the products (metabolites, endotoxins) released from them during growth. Periodontitis is a pathological inflammatory disease, as a response to years of prolonged exposure to a polymicrobial community in the gingival/periodontal pocket, involving tissue responses with alternating destructive (damage) and healing (repair) phases, but with a net loss of attachment and bone. The infectious nature of the microbes, and to what extent specific bacterial species, with specific virulence factors capable of interference with host defense and tissue repair, play in disease onset and progression, are still unclear. Most of the factors that used to be discussed in terms of virulence (proteases, LPS, invasive ability, fimbriae, capsule, leukotoxin) among microorganisms commonly associated with periodontitis should rather be termed survival factors, since they do not necessarily constitute factors for pathogenicity and damage but for the living, growth, and survival of the bacteria in deep, inflamed periodontal pockets. One exception is the leukotoxin produced by *A. actinomycetemcomitans* (and the JP2 genotype where clinical evidence support a causative role in disease progression) [101] and although the mechanism whereby the leukotoxin leads to tissue damage is still not known, this bacterial species best fulfils the designation of a putative periodontal pathogen in the human oral microbiota.

## Figures and Tables

**Figure 1 jcm-08-01339-f001:**
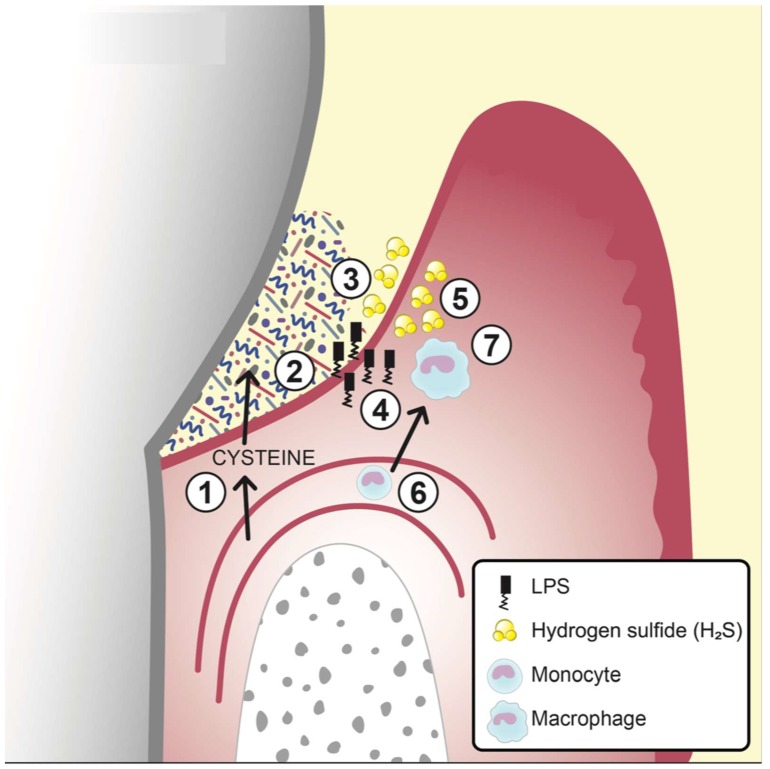
A schematic figure of an inflamed gingival pocket with subgingival plaque (biofilm) and the two signals that lead to the production and secretion of IL-1ß and IL-18 from monocytes/macrophages as reprinted from Basic et al. [59]. The numbers in the figure indicates the following: 1. Serum exudate from blood vessels containing serum proteins, peptides and amino acids including cysteine. 2. The exudate (gingival crevicular fluid, GCF) continues through the thin pocket epithelium (junctional epithelium) into the subgingival pocket. 3. The subgingival plaque, containing numerous, mainly Gram-negative, anaerobic bacteria with proteolytic capacity, degrade proteins, peptides and amino acids including cysteine. 4. Growing Gram-negative anaerobes release lipopolysaccharides (LPS) that penetrate the junctional epithelium into gingival connective tissues. 5. Growing Gram-negative anaerobes (*Fusobacterium* spp. *P. gingivalis*, *Treponema* spp., and others) produce metabolites e.g., hydrogen sulfide (H_2_S). 6. The inflammatory lesion attracts monocytes that migrate into the connective tissue and differentiate to macrophages. 7. The effect of LPS and H_2_S on macrophages and the subsequent production of the pro-inflammatory cytokines IL-1β and IL-18.

**Figure 2 jcm-08-01339-f002:**
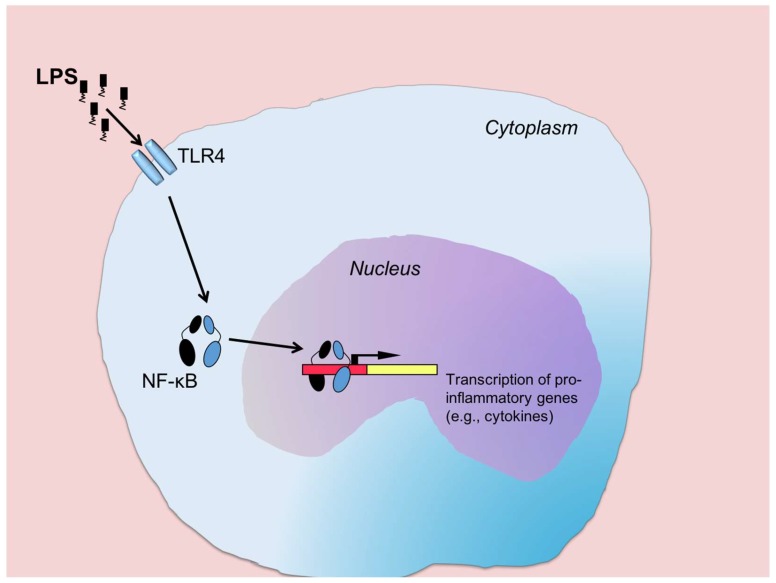
LPS from Gram-negative bacteria bind and activate TLR4 which leads to the activation and translocation from the cytoplasm of the transcription factor NF-κB. Inside the nucleus, NF-κB initiates the transcription of a wide variety of pro-inflammatory genes, e.g., those encoding for pro-inflammatory cytokines.

**Table 1 jcm-08-01339-t001:** Important characteristics for eight “putative periodontal pathogens” as described by Socransky et al. (1998) in colored complexes (as indicated in the table by the red and orange color) associated with periodontitis. *A. actinomycetemcomitans* (serotype b) did not fit into any of the complexes and is colored grey in the table.

“Putative Periodontal Pathogens”	Gram Stain	Main Metabolic Trait	Motility	Proteolytic Activity	Carbohydrate Fermentation	Major end Products	Factors of Significance	Subtyping
*Aggregatibacter* *actinomycetemcomitans*	Gramneg	Facultative anaerobic	No	Weak	Glycolytic	Lactic acid, Succinic acid Acetic acid, Propionic acid	Leukotoxin Cytodescen-ding (Cdt) toxin	Serotypes a-e, Non-serotypable isolates are frequent Specific genotypes (JP2)
*Porphyromonas* *gingivalis*	Gramneg	Anaerobic	No	Strong	Asaccharolytic	NH_3_ H_2_S Phenylacetic acid Indole	Gingipains (RgpA and Kgp)	FimA genotypes: I-V Arg-specific RgpA: A-C Lys-specific Kgp: I and II Capsular subtypes: K1-K6
*Tannerella* *forsythia*	Gramneg	Anaerobic	No	Strong	Weak glycolytic	H_2_S (weak) Acetic acid Propionic acid	Trypsin-like and PrtH proteases	Variations in the leucine-rich repeat BspA protein are existing but no subtyping is presented
*Treponema* *denticola*	Gramneg	Anaerobic	Strong	Strong	Weak glycolytic	H_2_S (strong)	Spirochetes (spiral shaped)	Seven oral Treponema species identified but no subtyping is known Hundreds of spirochetal genotypes (OTU′s) found
*Prevotella* *intermedia/* *nigrescens*	Gramneg	Anaerobic	No	Strong	Glycolytic	NH_3_ H_2_S Succinic acid Acetic	Indole	Capsule is produced but no subtyping is known
*Fusobacterium* *nucleatum*	Gramneg	Anaerobic	No	Strong	Glycolytic	NH_3_ H2S (strong) Butyric acid	Fusiform Morphology (threadlike)	Three subspecies reported variation in the outer-membrane structure
*Campylobacter* *rectus/gracilis*	Gramneg	Microaerophilic	weak	Weak	Asaccharolytic	H_2_S Succinic acid	Nd	not known
*Parvimonas* *micra*	Grampos	Anaerobic	No	Strong	Glycolytic	NH_3_ H_2_S (weak), Acetic acid	Nd	not known

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
