# Peer review of "Importance of Virulence Factors for the Persistence of Oral Bacteria in the Inflamed Gingival Crevice and in the Pathogenesis of Periodontal Disease"

_jcm, 2019, doi:10.3390/jcm8091339_

Round 1

Reviewer 1 Report

I do not have any additional comments

Reviewer 2 Report

I would like to congratulate the authors on the way they responded and corrected their manuscript.

This manuscript is a resubmission of an earlier submission. The following is a list of the peer review reports and author responses from that submission.

Round 1

Reviewer 1 Report

General comments

  This review focuses on the classical pathogenic factor of periodontal pathogens produced by putative periodontal pathogens and on bacterial metabolites that have been implicated in periodontal disease. 

  Among them, the authors look back on important reports in the past as a historical development of the pathogenesis of periodontal disease. Also, authors then touched on the process leading up to the polymicrobial synergy and dysbiosis hypothesis, the new pathogenesis developed recently.In addition, this manuscript introduces recent trends in the function of each pathogenic factor possessed by putative periodontal pathogens, and the periodontal pathogenicity of metabolites that have recently been attracting attention. However, “J. Clin. Med” does not dedicated to the field of oral microbiology, so this manuscript needs to be revised to attract researchers of general medical research. For instance, the relationship between systemic diseases and periodontal disease as described in section 9 would be better to added briefly in Abstract and Introduction section.

 In addition, some revisions shown as below should be necessary to forward this manuscript to consider for publication.

1.    L100-107: Enersen et al. have shown the virulence of fim A genotype II of P. gingivalisis stronger (Porphyromonas gingivalisfimbriae. Enersen M, Nakano K, Amano A. J Oral Microbiol.2013 May 6; 5). It should be better to add that for this manuscript.

2.    It overlaps with the point as mentioned above, FimA fimbriae of P. gingivalishas been recognized as a pathogenic factor involved in biofilm formation, invasion to gingival epithelial cells, and disruption of the immune response. Thus, a description for the function of the FimA fimbriae should be added in the manuscript.

3.    L427: In relation to periodontal pathogens and RA, the important paper on leukotoxin A of Aggregatibacter actinomycetemcomitans and citrullination of host proteins has reported (Aggregatibacter actinomycetemcomitans-induced hypercitrullination links periodontal infection to autoimmunity in rheumatoid arthritis. Konig et al. Sci Trans Med. 2016 Dec 14; 8 (369): 369ra 176). By including and citing this paper, the role and importance of leukotoxin would be further emphasized and this manuscript would be strengthened. 

4.    In the section “9. Systemic implications of periodontal bacteria”, the several papers for the relationship between periodontal disease and Alzheimer's disease have been published in recent years should be added in the manuscript. 

5.    More drawings/figures would be better to added in the manuscript, because there are some parts that are difficult to understand with sentences alone. For example, illustrations for the descriptions of LPS and TLR in “6.2. Endotoxins as pro-inflammatory and outer membrane vesicles” would be better to add. In addition, this manuscript would need a citing from a more recent paper.

Minor comments

1. L268 and L271: Legend for Figure 1; Please correct by making “2” of “H2S” as a subscript.

2. L267: Legend for Figure 1; “P. gingivalis” should be indicated in italics.

3. L363: Treponema denticolapossesses the dentilisin as a tissue destruction factor. Please add that in text and Table 1.

4. Table 1: Please correct by making 2 of H2S and 3 of NH3 a subscript.

5. L232 and Table 1: P. gingivaliscould produce butyric acid as well as Fusobacterium. Please add that in text and Table 1.

Reviewer 2 Report

This is a review article about the role of bacteria in periodontitis. The article aims to discuss the various microbial related factors that can contribute to periodontal disease. The article is very descriptive and adequately covers most of the major areas related to the topic.

Major Concerns:

1.     The title of the article is very confusing, its end with a question mark, however its unclear what the authors are questioning.

2.     Page 1 line 10: Claims oral infections are caused by low virulent organisms. This represents a limited view of oral infections.  While uncommon virulent organisms cause oral infections:
Example – HIV; Herpes, Strep. pyogenes, fungal infections etc.

3.     Page 1 line 13: Claims periodontitis is a result of prolonged physiological inflammation: This is not the current accepted paradigm.  Current view of Inflammation in context of periodontitis is that it is pathological and definitely not physiological.

4.     Page 2 line 59- 64; Is somewhat biased and perhaps incorrect interpretation of the literature; Several studies indicate a strong genetic component to etiology of periodontitis, as evidenced by lack of periodontitis despite poor oral hygiene in 10-15% of studied populations as well as the corollary of progression of disease despite excellent dental care in 10-15% of population in developed countries. Identification of familial aggregation in certain populations also support this view point.

5.     Page 5: line 116: the paragraph must  include a discussion of seminal work by Socransky in modifying Koch's postulate for the sake of completeness. Ref: Socransky S. S. Criteria for the infectious agents in dental caries and periodontal disease. J. Clin. Periodontol. 1979;6(Extra Issue):16–19

6.     Page 5: lines 135-138: The argument that gingivitis is physiological is akin to saying dermatitis or seborrhitis or rhinitis etc. is physiological. A physiological response must be result in resolution back to homeostasis, is the inflammation persists chronically that is not physiological.

Minor concerns:

1.     Page 1: line 38, 39. Suggest that the phrase ‘gingivitis in man’ should be included in single quotations.

2.     Page 16 Line 253-254: Figure is labeled Fig A which is misleading as its states below that as Figure 1.

3.     The language is dense and difficult to follow and some concepts are unnecessarily repeated in the article.

Reviewer 3 Report

How much could be influenced features of the dominating bacterial component of oral microbiota associated with periodontal inflammatory disease, e. g., plaque-induced gingivitis and periodontitis, by the contemporary presence of yeasts and viruses, including herpetic viruses and bacteriophages, in the subgingival dental plaque? Is it quotable in this review?

The quality and quantity of the immune responce resulting in the inflammation of periodontal tissue must be necessarily affected by various epigenetic factors. Is there a knowledge about these facts?